# A Rare Case of Patiromer Induced Hypercalcemia

**DOI:** 10.3390/jcm10163756

**Published:** 2021-08-23

**Authors:** Swetha Rani Kanduri, Kathryn J. Suchow, Juan Carlos Q. Velez

**Affiliations:** 1Department of Nephrology, Ochsner Clinic Foundation, New Orleans, LA 70121, USA; Kathryn.suchow@ochsner.org (K.J.S.); juancarlos.velez@ochsner.org (J.C.Q.V.); 2Ochsner Clinical School, The University of Queensland, Brisbane, QLD 4072, Australia

**Keywords:** patiromer acetate, hyperkalemia, hypercalcemia, hypomagnesemia, metabolic alkalosis

## Abstract

Patiromer is a calcium (Ca)-potassium (K) exchange resin approved for the treatment of hyperkalemia. Disorders of Ca or acid base balance were not reported in pre-approval clinical trials. We present a case of a patient with chronic kidney disease (CKD) with an unusual picture of hypercalcemia, metabolic alkalosis and hypokalemia upon intensification of patiromer dosing. A 56-year-old white man with CKD stage 4 (baseline creatinine 2.8 mg/dL) due to type 1 diabetes mellitus, proteinuria (1.5 g/g) and persistently high serum potassium 5.9 mEq/L attributed to type 4 renal tubular acidosis was evaluated in clinic. Due to high risk of CKD progression, patiromer 8.4 g daily, followed by 16.8 g daily was prescribed to enable renin angiotensin aldosterone system (RAAS) inhibitor. After 5 months of being on patiromer 16.8 g daily, routine laboratory tests revealed serum potassium 2.5 mEq/L, serum calcium 12.8 mg/dL and carbon dioxide 34 mEq/L. Patiromer was discontinued and thorough investigation held was negative for other causes of hypercalcemia. Five days after patiromer discontinuation, serum calcium returned to normal. The role of secondary hyperparathyroidism in this case remains unclear. We, therefore recommend cautious vigilance of patients receiving patiromer and undergoing dose escalation.

## 1. Introduction

Hyperkalemia is a common electrolyte abnormality in individuals with chronic kidney disease (CKD). If left untreated, hyperkalemia may be associated with significant morbidity and mortality [1]. As a result, treatment of hyperkalemia is of paramount importance for nephrology practitioners. Although diuretic therapy is generally the first line of therapy for chronic hyperkalemia, use of gastrointestinal (GI) methods to enhance elimination of potassium has also been a common practice. Among those GI approaches, clearly the use of oral sodium polysterene sulfonate (SPS) has historically been the most common therapy. However, undesired effects such as diarrhea and a purported greater risk for colonic perforations, its use has been questioned and reduced over the last decade [2]. Patiromer is a novel cation exchange resin approved by the FDA in 2015 for the chronic treatment of hyperkalemia [3,4,5]. Because it is better tolerated than SPS and its efficacy has been supported by data from randomized controlled trials, use of patiromer has gradually been adopted in current clinical practice. Additionally, multiple studies reported sustained potassium lowering effects and an increased fecal potassium excretion among maintenance hemodialysis patients [6,7].

While randomized control trials have reported adverse effects due to exposure to patiromer, such as hypomagnesemia and GI disturbances, there was no mention of hypercalcemia as a potential side effect in the package insert [3,4,8]. Herein, we report an unusual case of a white man with stage 4 proteinuric CKD who developed acute, severe hypercalcemia, in association with hypokalemia and metabolic alkalosis after few months of patiromer initiation.

## 2. Case Presentation

A 56-year-old Caucasian gentleman with CKD stage 4 (baseline serum creatinine 2.5 to 2.8 mg/dL, estimated glomerular filtration rate 26 mL/min) secondary to type 1 diabetes mellitus, sub-nephrotic proteinuria (urine protein-to-creatinine ratio (UPCR) 1.2–1.5 g/g), and essential hypertension presented for a routine office follow-up visit to a nephrology clinic. He had moderate persistent hyperkalemia (5.5–5.9 mEq/L) attributed to type 4 renal tubular acidosis associated with type 1 DM which had precluded initiation of renin-angiotensin system (RAS) blockade. Due to his high risk of progression to end-stage kidney disease, oral patiromer 8.4 g daily was prescribed with the intention of enabling initiation of RAS blockade when serum potassium reached consistently a range of approximately 4.0–4.5 mEq/L or lower. At that point, corrected calcium (corrected calcium to serum albumin) (Table 1) was 9.5 mg/dL, serum carbon dioxide were 24 mmol/L and serum parathyroid hormone (PTH) 338 pg/mL. Five months after starting patiromer 8.4 g daily, his serum potassium remained elevated at 5.0 mEq/L. The patiromer dosage was thus increased to 16.8 g daily. Three months after increasing the patiromer dose, the patient’s serum potassium was 4.1 mEq/L and hence, patiromer dose was maintained at 16.8 g. At that point, and RAS blocker irbesartan 75 mg daily was initiated. After 5 months of taking patiromer 16.8 g daily and 11 months of initial low-dose patiromer initiation, the patient developed notable electrolyte derangements including potassium 2.5–3.5 mEq/L, carbon dioxide 30–34 mEq/L, calcium 10.7–12.8 mg/dL and serum creatinine of 2.7–2.9 mg/dL over a span of 3 months. These laboratory abnormalities were noted upon routine laboratory testing but not triggered by any specific symptomatology consistent with hypercalcemia including weakness, confusion, nausea, vomiting or constipation. The patient denied use of recreational drugs, over-the-counter medications, thiazides, calcium supplements or vitamins. A renal ultrasound obtained four years prior to this presentation revealed increased echogenicity bilaterally and no evidence of hydronephrosis or stones. A new renal ultrasound was not performed. Extensive investigation for etiology of hypercalcemia did not reveal any potential explanation. Laboratory work-up included measurement of parathyroid related peptide (PTHrP), 1,25-OH-vitamin D, 25-OH-vitamin D, thyroid stimulating hormone (TSH), serum protein electrophoresis (SPEP) with immunofixation (IFE), serum free light chain ratio (sFLC), cortisol, and adrenocorticotropic hormone (ACTH) and were all unremarkable. Serum PTH levels were 21 pg/mL and urinary calcium was 4.4 mg/dL (normal 0–15 mg/dL). Serum aldosterone and plasma renin activity obtained to rule out hyperaldosteronism and mineralocorticoid derangements, returned to be within normal limits. When the serum calcium reached its highest value at 12.8 mg/dL, serum potassium at 2.5 mEq/L and serum creatinine at 2.9 mg/dl, patiromer was immediately discontinued. Five days after discontinuation of patiromer, hypokalemia, hypercalcemia and acute kidney injury (AKI) resolved, with repeat labs showing serum potassium of 4.1 mEq/L, serum calcium of 8 mg/dL, serum creatinine of 2.3 mg/dL and eGFR of 30.6 mL/min. The metabolic alkalosis persisted 3 days longer than the hypokalemia and hypercalcemia, with serum carbon dioxide ranging 32–36 mEq/L (Table 1). However, review of labs drawn 3 and 6 months after patiromer was discontinued showed improved metabolic alkalosis, with serum carbon dioxide ranging 27–29 mEq/L, while serum potassium ranged from 5.1–5.5 mEq/L and serum calcium ranged from 9.6–9.8 mg/dL. Because of the timing of hypercalcemia, hypokalemia and AKI respect to the exposure to patiromer and an almost immediate resolution of the abnormalities after discontinuation of patiromer, it was concluded that patiromer was the most likely culprit.

## 3. Discussion

Patiromer is a non-absorbable, sodium-free, calcium-based polymer that enables the use of RAS blockade in patients with CKD or heart failure. Patiromer has a rapid onset of action of 5–7 h and is considered an efficacious and safe method of lowering serum potassium in hyperkalemic patients [1].

The effective potassium-lowering capacity of patiromer has resulted in its widespread use in patients with advanced kidney disease, diabetes, and congestive heart failure [3,4,9]. However, an excess load of calcium may be delivered in exchange for the potassium as an 8.4 g dose of patiromer contains 1.6 g of calcium [10]. Notably, hypercalcemia secondary to patiromer has been only rarely reported to date [11,12]. To our knowledge, this case is the third report of patiromer-induced hypercalcemia. In addition, our case is novel because of the concomitant presence of metabolic alkalosis and hypokalemia that accompanied the increase in serum calcium. 

Calcium released from patiromer is reabsorbed by the gastrointestinal tract and enters systemic circulation where it can be renally excreted or remain in circulation. Theoretically, excess calcium has various plights, each potentially associated with different adverse effects. Calcium can bind anions such as oxalate and phosphate, which are excreted in the GI tract, but in the kidneys may deposit to form calcium stones (Figure 1). Calcium that remains in systemic circulation eventually can deposit in tissues, leading to calciphylaxis [13]. Hypercalcemia was not observed in the clinical trials that assessed the efficacy and safety of patiromer. Additionally, in post hoc efficacy and safety analysis of the TOURMALINE study [14], serum calcium levels remained unchanged in patients with early CKD. Increased intestinal absorption of calcium results in hypercalciuria as reported by Bushinsky et al. in healthy volunteers [15]. Interestingly, our patient developed worsening hypercalcemia (from 10.7 to 12.8 mg/dL) over the course of 5 months after the escalation of patiromer dosage, suggesting a subacute rise in serum calcium levels. Patiromer-associated hypercalcemia in our patient could be attributed to the combination of advanced CKD and diminished urinary calcium excretion. Although not completely clear, the fact that the PTH level was not fully suppressed in the setting of elevated serum calcium levels suggests the possibility of underlying secondary hyperparathyroidism playing a role in the observed hypercalcemia. Presence of co-existing conditions namely multiple myeloma, granulomatous disorders or malignancies might increase the occurrence of hypercalcemia among patients with advanced CKD.

Along with hypercalcemia, the patient developed concomitant hypokalemia. Undoubtedly, patiromer-induced GI elimination of potassium could have played a significant role in inducing severe hypokalemia in this patient. Furthermore, development of hypokalemia is a known phenomenon in the context of acute hypercalcemia. Hypercalcemia can mimic the effect of loop diuretics. Increase in filtered calcium leads to increased delivery of calcium to the medullary thick ascending loop of Henle (mTALH) that results in increased paracellular reabsorption of calcium. There, detection of calcium by the calcium sensing receptor (CaSR) localized in the basolateral side of the mTALH leads to inhibition of the renal outer medullary potassium channel (ROMK) thereby abrogating the function of the sodium-potassium-2 chloride co-transporter (NKCC2) [16]. This mechanism results in reduction of sodium reabsorption in the mTALH and consequently increased delivery of sodium to the collecting duct where sodium is exchanged for potassium, ultimately resulting in increased kaliuresis. In addition to the loop diuretic-like effect, hypercalcemia also leads to reduced insertion of aquaporin 2 (AQP2) water channels, nephrogenic diabetes insipidus and polyuria. Polyuric states can independently increase kaliuresis by activation of the flow-dependent BK channels, increased luminal chloride and sodium, kaliuresis, and a diuretic-like effect (Figure 1). Moreover, the diuretic effect and renal arteriolar vasoconstriction mediated by hypercalcemia can precipitate AKI episode as encountered in our patient. Increased urinary sodium delivery to the distal convoluted tubule and a volume contracted state results in subsequent release of aldosterone, resulting in hypokalemia and concomitant metabolic alkalosis.

The development of metabolic alkalosis seen in this case is not unexpected in the context of hypokalemia. Although the diagnosis of metabolic alkalosis ultimately requires a blood gas analysis for confirmation, it is cumbersome to obtain either an arterial or a venous blood gas sample in the outpatient setting. In addition, the co-existence of hypercalcemia and hypokalemia at presentation and the subsequent resolution of elevated carbon dioxide levels following normalization of serum calcium and serum potassium suggests that metabolic alkalosis was the most likely acid-base disorder rather than respiratory acidosis. Of note, patient did not have a diagnosis of any pulmonary condition. Hypokalemia activates the hydrogen-potassium ATPase (H^+^-K^+^-ATPase) in the collecting duct resulting in reclamation of potassium at the expense of H^+^ loss, i.e., development of metabolic alkalosis. While patiromer primarily binds potassium, to a lesser extent, patiromer also binds other cations, such as hydrogen [17]. Thus, loss of hydrogen ions bound to patiromer resin in the GI tract may have also contributed to the metabolic alkalosis observed in this patient.

## 4. Conclusions

Although the reported adverse event profile of patiromer is largely benign, acute hypercalcemia can rarely occur. As in our case, hypercalcemia may be associated with hypokalemia and metabolic alkalosis and may manifest several months after patiromer initiation. This highlights the importance of careful, frequent monitoring of serum chemistries for the duration of patiromer use. In patients with advanced CKD who may have blunted urinary calcium excretion, caution should be exercised when initiating patiromer or increasing its dose.

## Figures and Tables

**Figure 1 jcm-10-03756-f001:**
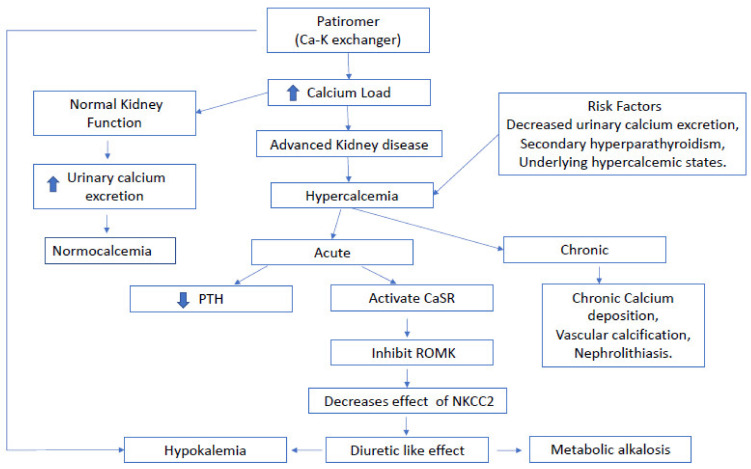
Schematic representation of proposed acute and chronic calcium homeostasis in patients on patiromer with advanced kidney disease. PTH: parathyroid hormone. ROMK: the renal outer medullary potassium channel. CaSR: the calcium sensing receptor. NKCC2: the sodium-potassium-2 chloride co-transporter.

**Table 1 jcm-10-03756-t001:** Laboratory values of the patient before initiation of patiromer, during and after discontinuation of patiromer.

Laboratory Parameter	Normal Range	Before Patiromer Initiation	Laboratory Values while on Patiromer	Laboratory Values Immediately after Discontinuation	Five Days after Discontinuation
Sodium (mEq/L)	136–145	138	141	141	141
Potassium (mEq/L)	3.4–5.1	5.7	2.5	2.5	4.1
Carbon dioxide (mEq/L)	22–29	24	34	32	32
Chloride (mEq/L)	96–105	105	97	98	103
Calcium (mg/dL)	8.6–10.2	9.3	12.6	11.2	7.7
Corrected calcium (mg/dL) ^	8.6–10.2	9.5	12.8	11.4	8
Albumin (g/dL)	3.8–4.5	3.8	3.7	3.7	3.6
Phosphorous (mg/dL)	2.5–4.5	3.1	2.3	2.4	2.5
Creatinine (mg/dL)	0.5–1.1	2.7	2.9	2.8	2.3
eGFR (ml/min/1.73m^2^)	120	26	* N/A	* N/A	30.6
Alkaline phosphatase (U/L)	35–104	66	39	39	39
PTH (pg/mL)	15–72	338	21		
PTHrP (pm/L)	<4	* N/A	1.2		
25-Hydroxyvitamin D (pg/mL)	30–100	20	14		
1,25-Hydroxyvitamin D (pg/mL)	19.9–79.2	25	24		
Urinary calcium (mg/dL)	0–15	* N/A	4.4		

* N/A: not applicable, ^ Corrected calcium: corrected calcium for serum albumin (corrected calcium (mg/dL) = measured total calcium (mg/dL) + 0.8(4-serum albumin(g/dL); PTH: parathyroid hormone, PTHrP: parathyroid hormone-related peptide.

## Data Availability

Not applicable.

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
