# Peer review of "A Rare Case of Patiromer Induced Hypercalcemia"

_jcm, 2021, doi:10.3390/jcm10163756_

Round 1

Reviewer 1 Report

In this manuscript the authors present the third case reported in the literature of hypercalcemia associated to patiromer. Although this topic is not new, the presence of metabolic alkalosis and hipokaliemia is interesting, and the explanation of the pathogenesis of thins phenomena well described.

Reviewer 2 Report

Hyperkalemia is common in patients with chronic kidney disease (CKD) and can cause life-threatening side effects in maintenance hemodialysis patients. Kanduri S.R. et al. present a case of patiromer-induced hypercalcemia in a CKD patient with type 1 diabetes mellitus. Data on patiromer treatment in CKD patients are rare, thus it is important to raise awareness of potential adverse effects of patiromer treatment similar to ion-exchange resins like sevelamer, polystyrene sulfonate and cholestyramine.

Major comments:

  • The management of hyperkalemia in CKD patients or end-stage kidney disease on hemodialysis using cation-exchange resins e.g. polystyrene sulfonate can cause gastrointestinal complications such as constipation, diarrhea and nausea, vomiting, abdominal discomfort. Did patiromer induce any treatment-related adverse events in this patient?
  • The authors wrote in line 68-69 “These laboratory abnormalities were noted upon routine laboratory testing but not triggered by any specific symptomatology”. What do the authors mean by “specific symptomatology”? Possibly gastrointestinal complications? This should be specified in the text.
  • eGFR values not on at baseline but also during patiromer treatment, immediately after discontinuation and five days after discontinuation should be measured and included.
  • What do the authors mean by “renal ultrasound was unremarkable”? Please specify. An image of a CT scan or ultrasound image of the kidneys could be included. Although ultrasound did not show any abnormalities, this does not exclude that patiromer did not induce further kidney injury, which according to the serum creatinine and urea concentration seems to be the case. Thus, in the case kidney biopsies of this patient have been performed, it would be important to perform PAS or H&E staining as well as aquaporin 2 expression.     
  • In a recent study with 27 anuric patients with hyperkalemia, the authors observed not only changes in serum potassium, sodium, and calcium following patiromer treatment but also in magnesium (PMID 32750456). I just wonder how the levels might be in the patient presented here. Hypomagnesemia?
  • Might be worth discussing data on patiromer treatment in CKD patients receiving hemodialysis (PMID: 32750456, 27784004, 34327218, 28122118) versus without. Several studies are available, which are not included in the reference list.
  • How do the authors define metabolic alkalosis? Because an increased serum bicarbonate level does not ultimately mean that this patient has metabolic alkalosis? Have the authors also looked at other parameters including an arterial blood gas test?
  • In my opinion, the authors have not fully addressed whether this patient has an acute or chronic hypercalcemia by using only laboratory parameters.

Minor comments:

  • Please clarify what “corrected calcium” level in Table 1 means and how it was calculated.
  • What does “N/A” stand for in Table 1?

The figure legend should be changed to “schematic representation with proposed …” because the authors have not looked at the physiology and pathophysiology on calcium homeostasis in the kidney in detail. 

Author Response

Reviewer 2

We appreciate the valuable suggestions and recommendations by the Reviewer. We believe that, as a result of the input provided, our manuscript has gained more merit. We have revised the manuscript accordingly.

1) Reviewer’s comments: The management of hyperkalemia in CKD patients or end-stage kidney disease on hemodialysis using cation-exchange resins e.g., polystyrene sulfonate can cause gastrointestinal complications such as constipation, diarrhea and nausea, vomiting, abdominal discomfort. Did patiromer induce any treatment-related adverse events in this patient?

Response: Thank you for the comment. While patiromer has been associated with gastrointestinal side effects, most patients tolerate it well. Our patient did not experience nausea, vomiting, abdominal discomfort, diarrhea or any other gastrointestinal symptom.

2) Reviewer’s comments: The authors wrote in line 68-69 “These laboratory abnormalities were noted upon routine laboratory testing but not triggered by any specific symptomatology”. What do the authors mean by “specific symptomatology”? Possibly gastrointestinal complications? This should be specified in the text.

Response: We appreciate the comment. Our patient did not manifest any specific symptomatology consistent with hypercalcemia including weakness, confusion, nausea, vomiting or constipation. We included the above statement in manuscript in the “Case Presentation” section.

3) Reviewer’s comments: eGFR values not on at baseline but also during patiromer treatment, immediately after discontinuation and five days after discontinuation should be measured and included

Response: We appreciate the suggestion and are thankful for bringing up a valuable point. The patient sustained an AKI episode which became very evident after the improvement in serum creatinine from 2.9 to 2.3 mg/dL following the discontinuation of patiromer. The episode of AKI was most likely due to hypercalcemia. This is supported by rapid resolution of AKI upon normalization of serum calcium level. Hypercalcemia is a well-established cause of AKI by two mechanisms: renal arteriolar vasoconstriction and volume depletion via a loop diuretic – like effect on the Na-K- 2Cl cotransporter localized in the medullary thick ascending loop of Henle.  We reported eGFR (26 ml/min) at baseline and have now added a value after discontinuation of patiromer (eGFR increased to 30.6 ml/min). However, we chose not to apply eGFR values during the AKI episode while on patiromer because it is not recommended to utilize estimates of GFR during AKI due to lack of a steady state. We have added eGFR values in “Case Presentation” section and in Table 1 in manuscript

4) Reviewer’s comments: What do the authors mean by “renal ultrasound was unremarkable”? Please specify. An image of a CT scan or ultrasound image of the kidneys could be included. Although ultrasound did not show any abnormalities, this does not exclude that patiromer did not induce further kidney injury, which according to the serum creatinine and urea concentration seems to be the case. Thus, in the case kidney biopsies of this patient have been performed, it would be important to perform PAS or H&E staining as well as aquaporin 2 expression

Response: We thank the Reviewer for the important question. A renal ultrasound obtained 4 years prior to this presentation revealed increased echogenicity bilaterally and no evidence of hydronephrosis or stones. A new renal ultrasound was not performed because of the high suspicion for hypercalcemia-induced AKI.  We have included findings of renal ultrasound in manuscript in “Case Presentation” section indicating that it was not a recent exam. In regard to the need for a kidney biopsy, hypercalcemia, hypokalemia and AKI resolved almost immediately after discontinuation of patiromer. Therefore, a kidney biopsy was not indicated and was not performed in this case.

5) Reviewer’s comment: In a recent study with 27 anuric patients with hyperkalemia, the authors observed not only changes in serum potassium, sodium, and calcium following patiromer treatment but also in magnesium (PMID 32750456). I just wonder how the levels might be in the patient presented here. Hypomagnesemia?

Response:  We completely agree with the notion that patiromer can certainly bind to magnesium along with potassium, leading to hypomagnesemia as reported in many of the studies. Unfortunately, we do not have serum magnesium levels measured in this patient.

6) Reviewer’s comment: Might be worth discussing data on patiromer treatment in CKD patients receiving hemodialysis (PMID: 32750456, 27784004, 34327218, 28122118) versus without. Several studies are available, which are not included in the reference list.

Response: Thank you very much for providing relevant references. We added above mentioned citations to the manuscript. We also included the role of patiromer among hemodialysis patients in the manuscript.

7) Reviewer’s comment: How do the authors define metabolic alkalosis? Because an increased serum bicarbonate level does not ultimately mean that this patient has metabolic alkalosis? Have the authors also looked at other parameters including an arterial blood gas test?

Response: We thank the Reviewer for the valid concern. We agree that the diagnosis of metabolic alkalosis ultimately requires a blood gas analysis for confirmation. However, it is cumbersome to obtain arterial or venous blood gas samples in the outpatient setting. In our case, the co-existence of hypercalcemia and hypokalemia at presentation and the subsequent resolution of elevated serum carbon dioxide levels following normalization of serum calcium and serum potassium suggests that metabolic alkalosis was the most likely acid-base disorder rather than respiratory acidosis. Of note, patient did not have a diagnosis of any pulmonary condition. We have also added rationale to the Discussion section.

8) Reviewer’s comment In my opinion, the authors have not fully addressed whether this patient has an acute or chronic hypercalcemia by using only laboratory parameters.

Response: Our patient developed worsening hypercalcemia (from 10.7 to 12.8 mg/dL) over the course of 5 months after the escalation of patiromer dosage, suggesting a subacute rise in serum calcium levels. We have added this qualifier to the Discussion section.

Minor Comments:

1) Reviewer’s comment: Please clarify what “corrected calcium” level in Table 1 means and how it was calculated.

Response: “Corrected calcium” refers to the value corrected for serum albumin concentration. We have included in manuscript and the footnote of Table 1.

2) Reviewer’s comment What does “N/A” stand for in Table 1?

Response: N/A represent “not applicable”. It is now mentioned in the footnote of Table 1

3) Reviewer’s comment; The figure legend should be changed to “schematic representation with proposed …” because the authors have not looked at the physiology and pathophysiology on calcium homeostasis in the kidney in detail. 

Response: We have modified the figure legend accordingly

Round 2

Reviewer 2 Report

The authors have addressed all concern.